# Differential Tree Growth Response to Management History and Climate in Multi-Aged Stands of *Pinus pinea* L.

**DOI:** 10.3390/plants13010061

**Published:** 2023-12-23

**Authors:** Vittorio Garfì, Giuseppe Garfì

**Affiliations:** 1Department of Bioscience and Territory, University of Molise, c/da Fonte Lappone, I-86090 Pesche, Italy; vittorio.garfi@unimol.it; 2Institute of Biosciences and BioResources, National Research Council, Via Ugo la Malfa, 153, I-90146 Palermo, Italy

**Keywords:** age classes, dendroecology, growth dynamics, spatial patterns, thinning, tree competition, water table

## Abstract

The possible differential response to the climatic fluctuations of co-occurring trees of different ages is still poorly known and rather controversial. Moreover, in managed forests, such a picture is further complicated by the impact of silvicultural practices. With this concern, in a multi-aged umbrella pine stand in the Maremma Regional Park (Tuscany, Italy), the spatial patterns and tree-ring response to the climate were investigated by differentiating trees into three classes, i.e., young, mature, and old. The aim was to assess the role of past management in shaping the current stand structure and affecting the growth dynamics at different ages, as well as to evaluate the possible shifting of tree adaptation to the climatic variables throughout plant aging. Our outcomes proved that the current mosaic of even-aged small patches results from a multifaceted forest management history. Until the 1960s, silvicultural treatments seemed more suitable in promoting tree growth and regeneration. Later on, inappropriate and/or untimely thinning probably triggered excessive competition from the top canopy trees, involving reduced stem and root system development in the younger plants living in the understory. Also, the intra-annual growth response to the climate showed some dependence on age. Younger trees are assumed not to be able to efficiently exploit water resources from the deep aquifer during the dry season, probably due to an insufficiently developed taproot, differently than older trees. Accordingly, appropriate and timely thinning, simulating frequent natural disturbances on small areas, could be a suitable management approach to promote sustained growth rates and regeneration processes, as well as healthy and vital trees at all life stages.

## 1. Introduction

Tree growth relies on a number of factors, such as climate, site quality, plant–plant interactions, genetics, age, and biotic disturbances (e.g., [1]), the latter including human impact and herbivory. The climate should, however, be regarded as a major driver [2]. Auxonomic dynamics, resulting from tree effectiveness in using trophic resources, is also controlled to some extent by endogenous variables linked to the individual ontogenetic stage of trees. In fact, it has been observed that in monospecific forest stands differing by social and/or age structure, the role of the climate on growth may vary significantly among trees belonging to diverse development phases [1,3,4,5,6,7,8].

However, in studies on tree-ring/climate relationships, it is usually recommended to select isolated or dominant trees in order to exclude *a priori* the effect of non-climatic factors and extract stronger climatic signals from tree rings [9]. According to this rule, also in investigations focusing on the age-dependent growth response of trees to the climate, isolated or dominant trees are mostly preferred in stratified sampling within uneven-aged stands (e.g., [5,10,11,12]). This approach allows for the prevention of finding biased climatic signal issuing from the competitive interaction by the canopy trees, as expected in suppressed or younger trees growing in the understory [2,13]. Yet, knowledge on such topics remains poor or controversial. Therefore, especially in managed forests, it might be of concern to assess the extent to which the climatic signal can affect growth in co-occurring young trees within multi-layered stands, and to evaluate, if any, its differential role with respect to the impact of forest management history.

Traditional forest management usually aims at optimizing the functionality and bioecological stability of tree stands, together with maximizing yield productivity. In most cases, priority is given to the cause–effect relationships between the spatial–temporal scheduling of silvicultural treatments and the growth response of trees through the control of intra- and inter-specific competition. Moreover, the intensity and spatial variability of thinning interventions and the resulting canopy gaps’ heterogeneity can significantly affect the success of regeneration processes, which in turn also depend on species-specific dispersal features [14]. This issue is of major importance for the maintenance of stable and functional stands, especially in close-to-nature silviculture, which also aims at the conservation of and/or the improvement in biodiversity [15]. In this regard, the analysis of spatial patterns at the stand level and among different stages of development has proved useful in providing insights into the impact of past forest management [10] and to eventually assist in modulating silvicultural treatments over time.

The umbrella pine (*Pinus pinea* L.) is one of the most iconic Mediterranean forest tree species. It occurs in native or naturalized stands all over Southern Europe, with a distribution area ranging from Portugal to some scattered stands around the Black Sea and the easternmost Mediterranean Sea coasts [16]. Since ancient times, this tree has been cultivated for protection purposes, mainly along the coasts, as well as for production reasons (timber and pine nuts) [17,18,19,20]. In the last few decades, umbrella pinewoods have been acquiring new functions related to their prominent landscape value and the more and more increasing demand for recreational uses, therefore also raising concern for conservation issues. Accordingly, the coastal stands have been included among the habitats of priority interests (2270—Wooded dunes with *Pinus pinea* and/or *Pinus pinaster*) by the ‘Habitats’ Directive 92/43 EEC. Largely originated from afforestation, the silvicultural treatments for this shade-intolerant species have usually been based on clear-cutting followed by planting or sowing [17,21,22], resulting most often in even-aged structures.

With reference to these topics, an investigation was carried out in the Maremma Regional Park (Tuscany, Italy), where a mosaic of structural types of umbrella pinewoods, from mono- to multi-layered stands, occurs [23,24]. The study focused on the auxonomic dynamics of a multi-layered forest patch, separately analyzing three co-occurring development stages, i.e., young (Y), mature (M), and old (O) trees. The aim was to (i) evaluate how past silvicultural treatments could have contributed to shape the current spatial patterns; (ii) elucidate which implications the management history could have had in terms of the differential growth of the three distinct development stages; and (iii) analyze the age-dependent growth response to the climate, with inferences concerning the possible shifting of tree adaptation to the environment throughout plant aging.

## 2. Study Area and Management History

The study site lies within the Pineta Granducale di Alberese, a pinewood area of 588 ha [25], which represents a quite distinctive forest, landscape, and naturalistic land unit. This is why, since the 1970s, the Pineta Granducale di Alberese has been placed under a protection regime, with inclusion in the Maremma Regional Park, and the subsequent recognition as a Natura 2000 Site (SAC IT51A0014—Pineta Granducale dell’Uccellina).

*Pinus pinea* prevails on more than 75% of the lowland forest area, while the rest is dominated by maritime pine (*Pinus pinaster* Ait.) and, just on the coastline, by salt-tolerant and hygrophilous communities. In the understory, various Mediterranean shrub species abound, such as *Erica multiflora* L., *Pistacia lentiscus* L., *Juniperus* spp., *Rhamnus alaternus* L., *Myrtus communis* L., *Phillyrea angustifolia* L., *Cistus* spp., etc. [26].

The climate is typically Mediterranean. The average annual rainfall is 644.6 mm (period 1952–1998, weather station of Alberese–Casello idraulico), but it is extremely variable from year to year, both in terms of the amount and distribution. Precipitations concentrate in the autumn–winter months, with a maximum in November and a minimum in July. The average annual temperature is 14.5 °C, but the maximum exceeds 30 °C for about 35 days a year.

The geomorphology is characterized by a succession of low dune belts parallel to the coastline. The soils originated from alluvial deposits dating back to the Quaternary period [27] and consist mainly of shallow and poorly evolved regosols. The texture is sandy, with almost no structure and a very low water-retention and cation-exchange capacity. In the innermost part of the pinewood, within the inter-dune space, seasonal stagnant water bodies due to variations in the level of the water table can form in small depressions (the so-called “lame”). Depending on the rainfall patterns, the annual fluctuations of the water table can range from 60 to 200 cm from the ground level, with a minimum between August and September [28]. The recharge usually begins in October and can attain the maximum in any of the months between October and March, depending on the seasonal distribution and the amount of rainfall [29].

The Pineta Granducale di Alberese originated from an artificial afforestation started in the second half of the 18th century and continued until the first half of the 19th century as part of a vast program aiming to reclaim the marshy areas and to consolidate the dunes of the Alberese estate [30,31]. Subsequently, and up to the beginning of the 20th century, the pine forest spread further in the neighboring areas through natural dispersal, in addition to sporadic afforestation activities, until covering the whole current extent [31]. From the second half of the 19th century, the main forest products have been timber and fruit (pine nuts); free-range grazing by the Maremmana landrace cattle is also a typical traditional land use [23]. Since past time, pruning to maximize pine nut yields has been performed alongside thinning, in addition to the partial clearing of the shrubby understory in order to facilitate cattle grazing and promote pine recruitment. Pine nut gathering was performed annually on all plants aged from 15 to 80 years but reserving 10–15% of cones for dissemination.

Forest management has changed through time [21]. Initially, it consisted of selection cuts, which only involved the removal of dead, decaying, or unfruitful plants. After 1924, with the acquisition of ownership by the Opera Nazionale Combattenti (ONC), the silvicultural treatments, though based on the original scheme, turned to increasing intensity with the aim of reducing the density of older tree classes and favoring natural recruitment; thinnings were extended to overnumerary as well as to overgrown trees that could hinder regeneration, and concerned up to 300–400 large plants/year throughout the whole pinewood [21]. The treatment could be referred to as a selection-type cutting with a 6-year rotation interval, and since it was not always spatially uniform all over the entire pinewood [21,32], the result was a mosaic forest structure [23]. Even-aged stands, up to a few hectares large, alternated to bi- and multi-layered patches formed by a scattered canopy of older pines over younger tree cohorts [23], which on the whole shaped an uneven-aged-like forest across large sectors of the pinewood [25]. After inclusion in the Maremma Regional Park in 1975, forest management gradually shifted from production to primarily conservation and recreation purposes. During the last few decades, only damaged or dead pines have been removed, while pruning and pine nut gathering are still carried out.

## 3. Materials and Methods

### 3.1. Biometric Data and Spatial Patterns

The diameter at 1.30 m (DBH), the height (H), and the crown width projection (CW) along the four cardinal directions were measured in all pine trees with DBH ≥ 3 cm in a circular sample area of about 5000 m^2^ (radius 40 m), a size that is considered appropriate for investigations on small-scale spatial patterns [33]. In total, 111 trees were sampled and each of them was mapped using polar coordinates. Smaller pines less than 3 cm at DBH and with age ≥ 3 years were referred to as established regeneration (ER) and mapped, too. According to the criteria and procedures described later on (see Section 3.2 and Section 3.3 for details), large trees were grouped into three chronological classes: young (Y), mature (M), and old (O).

Map data were used to analyze the spatial patterns of the investigated stand through the univariate Ripley’s *K* function [34]. This function expresses the expected number of points (i.e., the trees) at an increasing 1 m step on the distance *d* from 1 to 40 m (i.e., half of the diameter length of our circular sample plot, as recommended to limit the edge effect). The *K* function quantifies the extent to which the tree distribution pattern is more or less dense than the random Poisson distribution as the distance *d* increases from each tree base. It is calculated from the equation:$$K\left(d\right)=A\sum_{i=1}^{n}\sum_{j=1}^{n}\frac{\delta_{ij}\;(d)}{n^{2}}\;\;\mathrm{for}\;i\neq j,$$
where *A* is the area of the plot, *d* is the distance interval, *n* is the number of trees, and *δ_ij_* is a counter variable that can range from 1 to 0 depending on *d* and the distance between trees *i* and *j*. In order to linearize *K*(*d*) and stabilize the variance, the square root transformation *L*(*d*) is usually used, as proposed by Besag [34]: $$L\left(d\right)=\sqrt{\frac{K(d)}{\pi }}-d$$

To assess the significance of deviation from random distribution, 99 Monte Carlo random simulations were generated, providing a 95% confidence envelope. *L*(*d*) values within the envelope indicate a random pattern, whereas *L*(*d*) above or below the envelope indicates significant clustering or regular patterns, respectively. Spatial patterns were analyzed for all trees of the sample plot, and separately for each of the age classes Y, M, and O, and for ER as well.

Also, the spatial relationships by means of a two-by-two comparison between the Y, M, and O trees were investigated over a distance from 1 to 40 m by the bivariate *K*_12_(*d*) function, which is a generalization of the *K*(*d*) function [35]. Based on the assumption that trees of the Y, M, and O classes are sexually mature and able to disperse, the inter-class relationship was also analyzed between ER and all adult trees taken together in order to evaluate the spatial patterns of the regeneration processes throughout the most recent period. The bivariate *K*_12_(*d*) function calculates the spatial interactions between sub-populations (i.e., the age groups) within the same plot, following the equation:$$K_{12}\left(d\right)=\frac{n_{2}K_{12}\left(d\right)+n_{1}K_{21}(d)}{n_{1}+n_{2}}$$
where *n*_1_ and *n*_2_ are the number of trees of age class 1 and 2, respectively. The same square root transformation as above was applied to obtain:$$L_{12}\left(d\right)=\sqrt{\frac{K_{12}(d)}{\pi }}-d$$

The significance of deviation from the null hypothesis of spatial independence between age groups was tested by adopting a 95% confidence envelope from the toroidal shift null model as suggested by Wiegand and Moloney [36]. In the case of spatial independence between pine group 1 and pine group 2, *L*_12_(*d*) is within the confidence envelope, whereas *L*_12_(*d*) is above or below the confidence envelope, respectively, in the case of a significant positive (attraction) or negative association (repulsion) between pine group 1 and pine group 2. Attraction and repulsion are defined as the tendency of the trees in the two groups to be, respectively, closer or further apart than they would be if they were distributed independently of each other. Univariate and bivariate functions were calculated by using the software package Programita (v2018) [36,37].

### 3.2. Tree-Ring Data and Age Structure

Two cores were extracted from each sampled tree at DBH using a Pressler’s borer on the cross sides of the trunk, for a total of 222 cores. The samples were subsequently prepared according to classical procedures [38] and the ring widths were measured at an accuracy of 0.01 mm by a computer-linked mechanical platform (LINTAB 6) under a stereoscope and a software package (Time Series Analysis and Presentation, TSAP v4.89, Frank Rinn, Heidelberg, Germany). Cross-dating and correction of any measurement errors were carried out both visually (skeleton plot) and statistically, using the COFECHA program [39]; 26 cores were discarded at this step for the purposes of dendrochronological analyses, because they were poorly synchronized (correlation with the master chronology below 0.40) or undatable.

The individual chronologies were used to build up the age structure of the stand. Usually, the failure to intercept the pith for many cores and the unknown number of years required for plants of different ages to reach the sampling height can lead to biases in terms of accuracy. The first limitation was mitigated to some extent by estimating the number of rings missing from the pith by means of the pith locator method [40]. Subsequently, in order to estimate the actual age of the trees, a correction was made by assessing the number of years to reach the coring height. From an additional core extracted close to the ground on about 50% of the sampled trees, the difference in the ring number with respect to the coring height was calculated. The regression of these differences against the age at DBH provided the number of years to add to each ring series, which ranged between 15 and 3 from the youngest to the oldest trees, respectively. Then, the stand age structure was drawn by grouping plants into 5-year age classes [10,41]. Finally, using only the series that included the pith (25), the cumulative diameter inside bark (DIB) curves were plotted [11].

Concerning regeneration, the age of plants was assessed by counting either the branch whorls or, in a few cases of uncertainty, the tree rings from a stem disk taken at the tree base. Only plantlets ≥ 3 years old, in total 50 individuals, were considered as established regeneration (ER) [42] and included in further analyses.

### 3.3. Tree-Ring/Climate Relationships

In the purpose of dendrochronological analyses, the individual series from large trees were grouped in three chronological classes based on tree age at DBH: young (Y), up to 40 years old; mature (M), between 41 and 80 years; old (O), older than 80 years. The age breakdown for the different classes was aimed at obtaining either a sufficient length and representation of each class, or a good compromise between a reasonable disaggregation of the data, also possibly taking into account crucial moments of the stand management history (cf. [21]). In particular, (i) the age distribution patterns of all trees of the investigated stand; (ii) the age of “useful” fructification (over 40 years) with its implications in terms of the traditional function of pine nut production and the inherent silvicultural management (e.g., [1,17]); (iii) concerns related to the longevity of the species and the age of the oldest sampled individuals; (iv) dendrochronological issues, as already assessed for the species by Gadbin–Henry [43], were considered.

The analysis of the climate/growth relationship was carried out separately for each group. Within each class, in order to avoid any possible effect of the tree size on the growth responses to the climate, we discarded all individuals with stunted or irregular growth. The selection was made by including in each class only plants with at least two of the three main biometric parameters (DBH, H, CW) greater than the mean minus the standard deviation, calculated for groups of plants at 10-year intervals; in addition, only series counting more than 20 tree rings were retained. The 70 selected series were standardized by the ARSTAN program [44], using detrending methods that differed between the investigated age groups according to indications from the literature (e.g., [45,46]) and/or the greater significance of preliminary response functions. We used a cubic smoothing spline for older trees (class O), a negative exponential for class M, and an RCS (Regional Curve Standardization) function for class Y trees.

The impact of the climate on the radial growth of each of the tree groups was analyzed through response functions performed by the CALROB program (package PPPBASE) [47]. The program calculates an orthogonalized multiple regression using the bootstrap method [48], in our case based on 1000 random replications. The mean value of the multiple correlation coefficient gives a measure of the robustness of the climate/growth relationship, and the ratio between the verification correlation coefficient and its standard deviation (*r*/*s*) is considered significant at the 95% level when it is ≥1.96 in absolute value [2].

The dependent variable consisted of the standardized master series of each age class. The independent variables included 24 climatic regressors, where the 12 monthly precipitations (P) were associated with the 12 monthly maximum and minimum temperatures (P–Tmax and P–Tmin), separately. The climatic data were recorded at the weather station of Alberese–Casello idraulico (17 m a.s.l.) and, as is customary for studies on Mediterranean species, were organized according to the “biological year” [49], i.e., from October of the year preceding the growth (*t* − 1) to September of the year of growth (*t*). The time interval considered covered the common 40-year period (1958–1997) for all three chronological classes.

## 4. Results

### 4.1. Age, Tree Biometrics, and Spatial Structure

The age structure of the investigated stand shows two distinct plant groups, separated by a 30-year gap (Figure 1). The first group spans from 1857 to 1889 and includes the totality of trees in class O, aged 108 to 140 years. The second group is by far larger, representing almost 80% of all the sampled plants, and consists of M and Y individuals aged between 19 and 74 years. Their establishment occurred with no discontinuity between the mid-1920s and the early 1980s, following a modal distribution with a maximum around the early 1950s (about 50-year-old trees).

The main biometric parameters, namely the mean DBH, H, and CW, differ notably among the three age groups and, based on their respective coefficient of variation, revealed to be much more uniform in the O and M than in the Y trees (Table 1).

The grouping into the three age classes of Y, M, and O also mirrors some features of the stand spatial distribution, resulting in rather distinct patterns at the group level (Figure 2). The Ripley’s univariate analysis shows that throughout the whole plot, trees are regularly spaced at short distances (1–5 m), but cluster at distances between 10 and 27 m (Figure 3). At a more detailed scale, clumping especially concerns juveniles at a distance of 6–29 m, whereas mature trees show, on the contrary, independence at 1–6 m distance. For O trees, the *L*(*d*) values are always within the Monte Carlo 95% confidence envelope, indicating a random distribution at all spatial scales, with some clues of independence at the shortest distance. ER displayed a strong aggregation pattern at all distances up to 32 m. The bivariate analysis displays *L*_12_(*d*) values statistically significant only in the comparison between the M and O trees, suggesting repulsion between the two groups at distances from 2 to about 24 m (Figure 4). The associations between the Y trees and the two older groups shows a significant repulsion of Y vs. M at 1–5 m, but only clues of some negative spatial association with the oldest group at very short distances. Significant repulsion at the interval of 6–11 m is also shown between ER plantlets and all trees of the three age classes taken together.

### 4.2. Tree-Ring Data and Growth Features

The mean ring width, calculated over the entire length of each series, is smaller in the oldest trees with respect to the Y and M groups, attaining 92% and 65%, respectively (Table 2). Also, the typical initial effect of the age trend, which usually results in a greater ring width, drops rapidly in the Y trees, lasting only about two decades; conversely, it lasts up to more than 40 years in the M and O trees before achieving a stabilized growth (Figure 5). The cumulative DIB curves based on the 25 cores with pith, compared at the same cambial age, confirm this pattern. The Y individual series are always below those of the two older tree classes (Figure 6), indicating that during the common age interval, the juveniles grew at a lower rate. A similar figure results in the cumulative DIB master curves obtained by averaging the individual series of each age class, aligned at the same cambial age after estimating the number of missing rings from the pith (Appendix A). Moreover, in this case, the M and O class curves are almost overlapping over the entire common period.

The descriptive statistics of the tree-ring width (Table 2) provide additional information about the series quality. The mean sensitivity, calculated from both the master series (MSm) and the average of the individual series (MSi), is globally high, as well as the cross-dating coefficient (CC) and the mean correlation with the master series (CM), indicating a remarkable sensitivity of the investigated stands to climatic fluctuations and a rather uniform response of trees within each age class. The first-order autocorrelation coefficient (AC_1_) was always higher than 0.61 (Y class) in the raw series, but autocorrelation was efficiently removed in all age classes after standardization.

### 4.3. Response Function Analysis

On the one hand, the response functions show that the climate explains a high variance of ring growth. In both the correlations with P–Tmax and P–Tmin, *R_V_* is never less than 0.73 (Table 2), with little differences according to age. On the other hand, the statistical significance, expressed by the ratio between the verification correlation coefficient to its standard deviation (*r*/*s*), shows a rising trend with increasing age. However, in the youngest chronology, *p* is not significant in the P–Tmin relationship, whereas it is <0.1 for P–Tmax. Only for the oldest trees (O) is *p* always smaller than 0.05.

The profile of the response functions globally shows the greater role of precipitation on the growth of the umbrella pine (Figure 7). Nevertheless, the different climatic parameters seem to perform differently depending on the plant age. In the P–Tmax relationship, the annual growth of the oldest trees is positively correlated to the autumn rainfall of October and November and prolongs until December for the M class. On the contrary, the Y plants appear to be sensitive to the amount of rainfall in December and early spring (March). Only for the M group does an inverse relationship between growth and rainfall also seem to occur in late summer. A very similar profile can be observed for the P–Tmin growth–climate response. Both the maximum and minimum temperatures have a positive influence in January, especially on the growth of age classes M and O, while only for the youngest trees does June’s maximum temperature play a positive role on growth.

## 5. Discussion

### 5.1. Management History as a Major Driver of Spatio–Temporal and Growth Dynamics

The main data related to the structural features of the investigated stands provide comprehensive evidence of the crucial role of silvicultural management and its changing schemes through time on tree growth dynamics. In the literature, with some exceptions (e.g., [42]), umbrella pinewoods are most commonly reported as even-aged (or mono-layered) stands [50,51] issuing from plantation or either natural or artificial regeneration following clear-cutting on more or less large surfaces [17,21]. Our uneven-aged studied stand therefore represents a valuable opportunity to improve knowledge about certain age-related processes (e.g., growth and regeneration; [42]) at the population level in multi-layered stands.

Each tree in a stand has an individual growth history depending on the different intensity, timing, and spacing of external disturbances [11], which lead to a multiform age and size structure. However, broadly speaking, some common traits can be identified in the case of tree groupings by selected criteria such as social position or age. In the investigated pinewood, despite the age distribution of the oldest tree cohort spans over about four decades, overall the O trees can be referred to as the last remnant of the vast plantations created from the first half of the 19th century onwards to reclaim the marshy areas and consolidate the dunes of the Alberese estate [31]. The current living plants, grown at the beginning within an even-aged stand type, are assumed to be the dominant or most productive individuals, and as such have been preserved from the periodic silvicultural or harvesting cuts applied through time. Accordingly, their growth during the early stages maintained a high rate since it has not been significantly, or at all, affected by inter-individual competition (Figure 6). Yet, their spatial patterns show a random distribution. In fact, any possible regular or geometric spacing, as is typical of artificial planting, is assumed to have been lost through time, since the preservation of the current oldest trees is mainly issued from unsystematic selection cuts especially targeting dead, decaying, or unfruitful plants [21]. This treatment, especially in the early decades after tree sexual maturity, somehow could have also allowed occasional regeneration processes, also considering that the young seedlings and saplings could benefit from the relative protection of the canopy cover against extreme solar irradiation [42]. Later on, recruitment appeared to decline until turning completely null between the mid-1890s and the mid-1920s (Figure 1), possibly arising from excessive canopy closure [21]. Tree crown development, not coupled to appropriate thinnings, involved increasing light competition from the top layer that hindered the regeneration processes of this shade-intolerant species [17].

The new management period, initiated by the ONC administration since the mid-1920s [30], prompted more regular and intensive thinnings, which favored new recruitment phases originating the current chronological M class. This is consistent with the spatial patterns of the M trees showing clear independence at a short distance (below 6 m) and some tendency to clumpedness around 16–19 m, as well as the evident relationship of independence (or repulsion) from the >80 years tree class at a short–medium scale (3 to 24 m). In fact, such a picture matches well with the clearings progressively created by the periodical silvicultural cuttings that allowed new generations’ establishment (Figure 2). Similar patterns have been reported for uneven-aged *P. pinea* stands in Spain [42], but also for other shade-intolerant species (e.g., *Larix decidua* Mill.), whose effective regeneration processes require medium to large size gaps [52].

As a matter of fact, the described spatial attributes seem to contrast with the barochoric dispersal mechanism of umbrella pine, because as reported by Masetti and Mencuccini [53], only 3% of produced nuts usually fall outside the crown area of the mother plant. Actually, the current M trees may grow quite far from their putative mother plants, but it must be considered that the gaps where they occur today may result from the progressive enlargement of clearings following the consecutive selection felling carried out during the ONC management phase within the O age group. Thinning concerning the M age class itself could have further impacted as well. Moreover, as reported by Rolando [54] in the investigated area, the facilitative role of birds cannot be excluded, especially the European jay (*Garrulus glandarius* L.), in both the clustered patterns and dispersal distance from the mother plants. As largely being known for bird-mediated dispersal interactions between *Quercus* sp. and jay [55], or *Pinus cembra* and nutcracker [10], it is possible that during summer–autumn, the jay feeds with pine nuts [54] and also places nuts in shallow caches in the ground or leaf litter, where they are protected from desiccation and consumption by other predators. Nuts that remain unrecovered could contribute to pine regeneration. In addition to the successful establishment, it can be assumed that in appropriate openings, the M age class enjoyed a relatively low competitive interaction, which favored a faster growth and stimulated stems to approach the canopy at a younger age and with a larger diameter [56]. As a result, the growth rate of both the M and O trees revealed to be comparable at the same cambial age (Appendix A).

The spatial and growth patterns of the Y class depict different tree life-history traits. Though at the stand level juveniles show a significant clustering at intermediate distances (6–29 m), the bivariate analysis indicates no evident preferential establishment in the present open spaces, as displayed in the case of the M age group. A tendency to repulsion is observed at a short distance (<5 m) from the two oldest age classes, which is, however, significant only in the association with the M trees. Such spatial attributes are somehow in agreement with the average crown width of the two upper layers, whose radius varies between around 4 and 5.2 m for the M and O trees, respectively. In an investigation about the association relationships between crown patterns and recruitment, Barbeito et al. [42] showed that in both even- and uneven-aged umbrella pine stands, the aggregated regeneration pattern is a rule. These authors also highlighted that the spatial positive association of regeneration is stronger with older, large crown trees, that is, the greater fruit producers. At Alberese, the umbrella pine has been specifically bred for such purposes [21] and until the 1970s, trees have been regularly thinned and pruned to promote crown enlargement and seed crop production. In the present multi-layered stand, it can be inferred that the current O and M trees have remained, for all along their life cycle, the most fruitful individuals, therefore largely contributing to dispersal and regeneration processes. The offspring, representing the present Y cohort, could, however, have established mainly in the neighborhoods but not below the crown of the putative mother plants. Furthermore, we can assume that at the time of their early life stage, the current Y trees found sufficiently large and suitable gaps, allowing their successful establishment. However, most probably, a progressive reduction in the open space size should have occurred later on, due to the crown expansion of upper layers coupled to the decrease in silvicultural practices, as a consequence of the new management policy related to the institution of the Maremma Regional Park in 1975 [23]. The spatial attributes of the ER cohort also mirror the impact of the protection policy on forest management and further emphasizes the picture already analyzed for the Y group. The strong clumping pattern corroborates the assumption of a significant reduction in the management intensity in the last few decades, especially involving an almost definitive stop in thinning. Also, the bivariate analyses address towards a preferred establishment of ER at a certain distance from existing adult stems in order to benefit from some suitable light conditions [42].

The new protection regime also involved hunting prohibition, which induced a fast increase in wild ungulate populations, mainly wild boar (*Sus scrofa* L.) and fallow deer (*Dama dama* L.). Their impact on pine seedlings/saplings due to browsing and trampling could have significantly affected the growth response and effectiveness of regeneration processes as well [57]. According to this scenario, the possible disturbance by ungulate browsing, in addition to the increasing light competition from the top layers, may explain the early slowing down of juvenile growth [56] (Figure 5), finally resulting in the remarkable lower growth rate of the Y group with respect to the oldest classes at a comparable cambial age (Figure 6).

Nevertheless, though the direct impact of management history could be considered the major driver of the depicted situation, such growth patterns might have been enhanced by additional external factors such as increasingly frequent extreme summer drought events [58,59]. Mazza and Manetti [12] and Cutini et al. [50] reported that in Central Italy a declining growth of umbrella pinewood was generally observed from the half of the 1970s onwards, due to a decreasing water supply depending by either the reduction in the rainfall trend or the impact of tourism and agriculture expansion on the depletion of stored soil water. The related lowering of the water table might have especially affected the growth of younger pines; in fact, unlike mature trees, which are provided with a deep taproot [60], their likely incompletely developed and shallower root system may not be able to access the deeper underground water resources, entailing a decline in their overall growth.

### 5.2. Growth and Climatic Response: Does Tree Age Matter?

Functional processes involved in tree growth are quite complex and diversified [1] and the issue of age-related growth relationships still remains one of the most debated topics (cf. [3,4,5,6,7,8,61,62]). In addition to physiological changes associated with aging [1,3], intra-population interactions (e.g., competition for light, water, and nutrients, especially in uneven-aged/multi-layered stands) play a basic role, which makes it difficult to separate the influence of internal from external drivers. In the present study, in addition to what was mentioned in the previous section, further clues to the impact of competition and the environment on tree growth are provided by the main dendrochronological data.

The mean sensitivity (MSm and MSi) was found to be globally high in all age classes; it is well above the critical threshold of 0.20 [2] and has a magnitude comparable with other umbrella pine populations from the Italian [12,29,59,63], Tunisian [64], and Greek [60] coastal areas. Such values suggest high sensitivity of tree-ring fluctuations to prevailing external factors such as the climate and, together with the high CC and CM coefficients, show that overall, the individual series respond uniformly to the climatic signal [2]. However, in contrast to what has been observed in other studies on age-related tree-ring/climate relationships, no clear upward or downward trend from younger to older classes has been detected. In studies on the growth rate and climate responses of *P. pinea* along central Italian coastal stands [12], and of *Larix decidua* and *Pinus cembra* L. in the Eastern Alps [5], MS was found to increase from young to old tree groups. Actually, the diverse sampling strategy between our survey and the two cited studies can be invoked to explain this discrepancy. In our case, we considered the totality of trees in the sample plot, irrespective of possible competition influences related to the individual social position, whereas in both the mentioned investigations, in order to avoid bias due to factors other than the climate, trees were selected among the dominant or isolated individuals. Accordingly, in our samples, we cannot exclude noising effects due to competitive interactions between tree layers, or even artefacts depending on the shortness (maximum 40 years) of the younger chronologies. Yet, the rather high auto-correlation coefficients should indicate in all age classes a remarkable influence of the previous year’s growth on the current year’s ring width, addressing to theoretically prefer residual modeling in the analysis of tree-ring/climate relationships. However, the preliminary response functions tested with various detrending methods, also including residual modeling, provided in the latter case less robust or mostly non-significant results. This is why indexed chronologies were ultimately chosen to evaluate the climate impact on the year-to-year growth fluctuations of the three pine age groups.

On the whole, the results of the response functions show that in all age classes, the role of the climate seems prominent in affecting tree growth responsiveness (*R_V_* between 0.73 and 0.80), albeit at different significance levels in the diverse tree groups. In particular, the Y trees were revealed to be less sensitive to the climate, and their response was barely significant at *p* < 0.1 (P–Tmax) or non-significant (P–Tmin). This picture was to some extent expected and, as discussed above, most likely quite depending on interclass competition effects, which can bias the climatic signal [5,10,11,12].

Nonetheless, the profile of the monthly response function is well consistent with the global processes analyzed at the entire stand scale. Autumn–winter and spring precipitation proved to be the main climatic drivers of the radial growth of *P. pinea* at Alberese, whereas temperatures have a secondary role. Interestingly, the intra-annual response to the climate showed some dependence on age; unlike the O and M groups, the Y class trees seem insensitive to autumn rainfall, and mostly dependent on the precipitation of early winter and spring. Age (or size-)-related differential response patterns were also observed in *Picea glauca* (Moench) Voss in subarctic Canada [3], as well as in *Larix decidua* and *Pinus cembra* in an Alpine environment [5], but it was not so evident in other *Pinus* species such as *P. halepensis* Mill., *P. pinaster*, *P. nigra* J.F. Arnold, and *P. uncinata* Ramond ex DC. stands [7,8,61]. With this concern, while taking into account the possible biases related to the social position of the sampled trees (isolated/dominant in the cited literature vs. spatially co-occurring in the present study), further aspects can be considered in addition to the issues discussed so far regarding competitive interactions. The positive influence of the precipitation of the rainiest months (October–November *t −* 1) on the growth of the two more aged pine groups is most likely associated with their key contribution to the replenishment of the deeper water table. Adult umbrella pines are provided with a multi-layered root system, with a deep and large-sized taproot [65], allowing trees to efficiently use water resources from different soil depths and sources such as freshwater and, above all, soil water and the water table [60,66]. The latter plays a crucial role in mitigating water stress and sustaining growth during the summer dry season in semi-arid habitats, especially considering that the low water-retention capacity of the Alberese sandy soil does not allow rainwater stratification in the upper layers, resulting in rapid drainage towards the greatest depths [51]. This explanation is consistent with the findings of many investigations on the tree-ring/climate relationships of umbrella pine carried out in other Italian coastal stands [12,29,50,51,60], as well as in Portugal [67], Tunisia [64], and Greece [60]. In contrast, in the younger class plants, the root system is assumed to have not attained full development and is still relatively shallow [6,8,65], so that their taproot is not yet long enough to reach the water table, which in the dry season can drop up to 2 m below the ground [28,65,68]. Accordingly, their growth can only depend on the freshwater input occurring in the months closest to or coinciding with the start of the growing processes, being indifferent to the more abundant autumn rainfall. The detrimental effect of the limited water supply could have been enhanced if we also consider the under-canopy precipitation flow, which can be lessened due to the excessive canopy density [62,69,70], as possibly resulting from irregular or decreasing silvicultural management after the 1970s. In addition to the above, in younger plants low growth rates can be further exacerbated in the event of consecutive years of low rainfall, as highlighted in recent research on the adaptation of *P. pinea* to climate change carried out in Greek and Italian pinewoods [60].

With respect to the temperature, once again, the M and O trees show a different behavior than the Y trees, proving to be sensitive to both the Tmax and Tmin in January. Liphschitz et al. [71] demonstrated under experimental conditions that the cambial growth of *P. pinea* is completely inactive during the winter months and resumes in April. However, as a rule in the Mediterranean area the water supply is not a limiting factor for growth at this time of the year, as also discussed above. Consequently, in case winter temperatures rise above the functional threshold for physiological processes, an early interruption of dormancy is possible, resulting in a longer growing season and wider tree rings. Such an explanation has also been suggested for umbrella pine in Sardinia [12], as well as for maritime pine in the Portuguese coastal stands [8].

A couple of monthly parameters, namely the negative correlation of September precipitation with the cambial activity of the M trees, and the positive influence of the June maximum temperature on the Y pines’ ring growth, remain difficult to interpret and are even the opposite from expected. With respect to the latter, a similar response was found in *Larix decidua* [5], thriving, however, in a quite different environment where the temperature is a major promoting factor for growth. This is not consistent with our case study, and since no further data have been found in the literature either on the significance of the negative impact of late-summer rainfall, any assumption can be biased.

Our findings support the idea that at the Alberese multi-layered pinewood growth response to the climate varies all along the life history of pine trees, resulting in the modulation of tree adaptation at the micro-environmental scale. The response to environmental conditions shifts with plant life stages not only according to age-related ontogenetic processes (e.g., increasing water-use efficiency) [6], but also owing to the increase in both the tree size and stand structural complexity. As shoot and root systems develop, plants progressively integrate into the environment, enhancing their fitness to small-scale fluctuations. With aging, trees are buffered against changes in the soil moisture because of their rooting patterns, and against changes in the temperature and nutrient availability due to their large above- and below-ground mass, enabling them to better access resources [72].

## 6. Conclusions

In the multi-aged umbrella pinewoods of Alberese, the presence of canopy gaps in the highest tree layers was revealed to be the key factor in driving the structural/demographic and growth patterns of stands. As a matter of fact, the current mosaic of variable size even-aged patches is the direct consequence of a multifaceted forest management history. Changes through time in silvicultural practices have resulted in greater or lesser ecosystem functionality depending on the timing, extent, and/or criteria of treatment. Accordingly, appropriate and timely thinning, simulating frequent natural disturbances on small areas as adopted in the ONC management phase, could be a suitable management approach to promote regeneration processes and sustained growth rates, as well as healthy and vital trees at all life stages. The soundness of such an assumption is also supported by research on old-growth forests [11,73,74]. These studies, albeit focusing on species other than the umbrella pine, emphasized the importance of both anthropogenic and non-anthropogenic disturbances on a variable spatial–temporal scale in achieving a functional steady state in forest ecosystems.

The effect of density modulation in reducing light competition and driving growth patterns of juvenile trees cannot be clearly separated from the influence of major environmental factors such as the climate [62]. As well as the above-ground biomass, the outcomes of our study suggest that inadequate silvicultural treatments can also result in the reduced development of the root system in younger plants living in the understory. Consequently, since they are not provided with a sufficiently deep taproot like older plants, they did not seem to be able to fully express their potential to adapt to the climatic fluctuations typical of the Mediterranean climate and local environment by efficiently exploiting water resources from the deep aquifer. Enlarging the canopy gap size could likely contribute towards mitigating this limitation. Moreover, the reduction in the high layer cover was revealed to be effective in improving the precipitation throughfall, increasing the amount of water reaching the topsoil [69], from which shallow-rooted younger trees can especially benefit.

Since years, many coastal umbrella pine stands such as the Alberese pinewood have largely been recognized as natural and cultural landscapes of high value. However, in the last few decades, their conservation is becoming more and more challenging, especially in the framework of the predicted climate changes, whose effects include a significant decrease in the precipitation amount. Therefore, adequate management strategies should be adopted and close-to-nature silviculture, which also aims at the conservation of and/or improvement in biodiversity, might be a valuable option.

## Figures and Tables

**Figure 1 plants-13-00061-f001:**
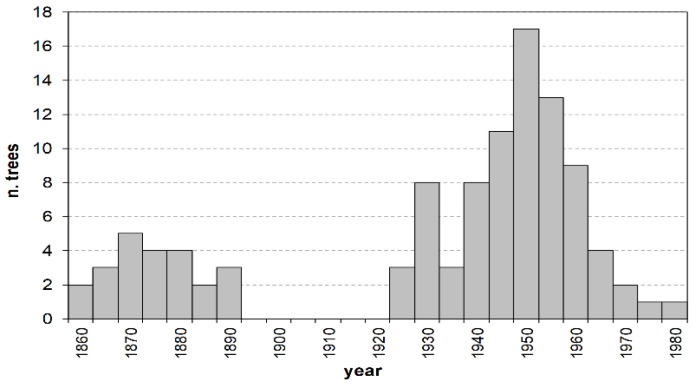
Age structure by 5-year age classes. Values on the x axis represent the age-class mid–point.

**Figure 2 plants-13-00061-f002:**
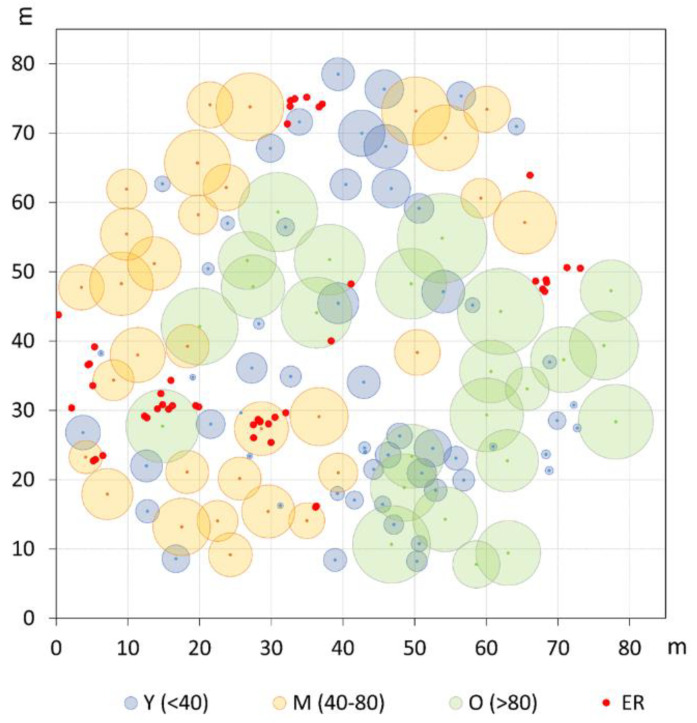
Spatial distribution and crown projection of old (O), mature (M), and young (Y) pine trees in the investigated plot. Established regeneration (ER) is represented by even-sized red dots.

**Figure 3 plants-13-00061-f003:**
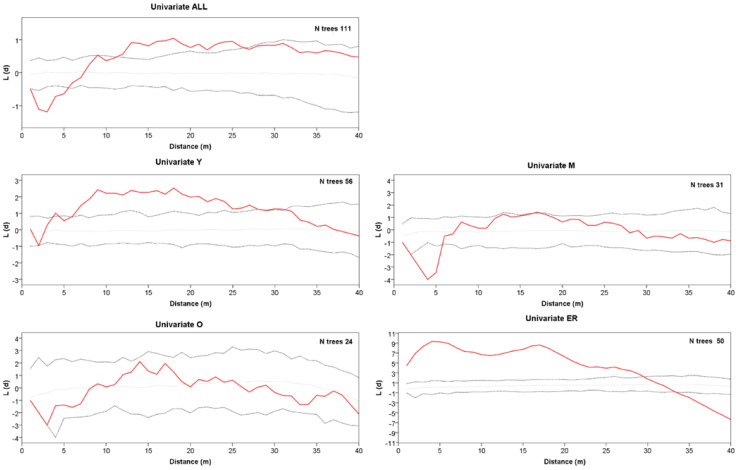
Univariate Ripley’s *L*(*d*) versus distance *d* for all trees, class Y, class M, class O, and ER cohort. The solid red line represents *L*(*d*). Dotted lines represent the Monte Carlo envelope constructed at 95% confidence level after 99 simulations. Positive and negative values *L*(*d*) indicate clumping or segregation, respectively.

**Figure 4 plants-13-00061-f004:**
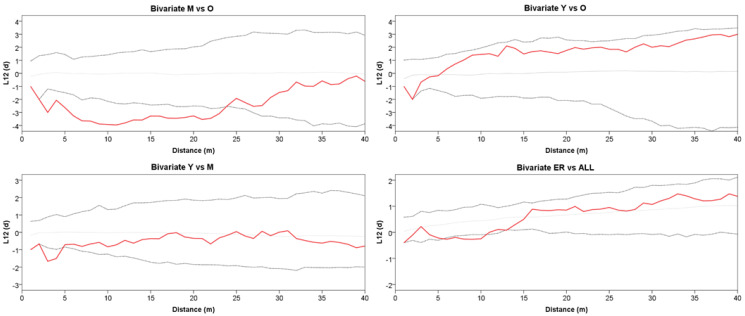
Bivariate Ripley’s *L*_12_(*d*) between two point patterns for Y vs. M, Y vs. O, M vs. O classes, and ER vs. all trees. The solid red line represents *L*_12_(*d*). The dotted lines represent the Monte Carlo envelope constructed at 95% confidence level after 99 simulations. Positive and negative values of *L*_12_(*d*) indicate attraction or repulsion, respectively.

**Figure 5 plants-13-00061-f005:**
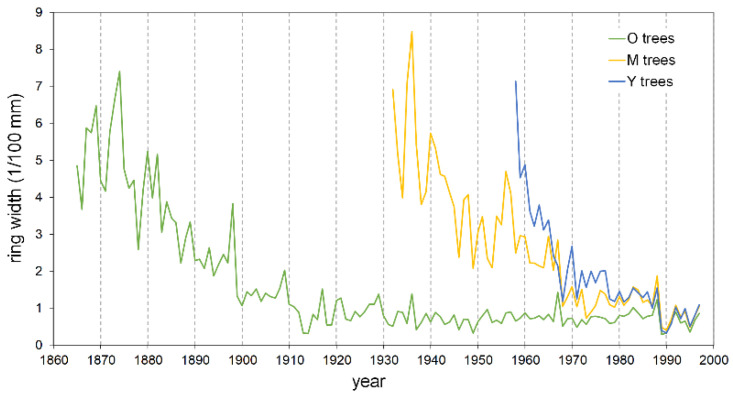
Master raw tree-ring chronologies of young (Y), mature (M), and old (O) pine trees.

**Figure 6 plants-13-00061-f006:**
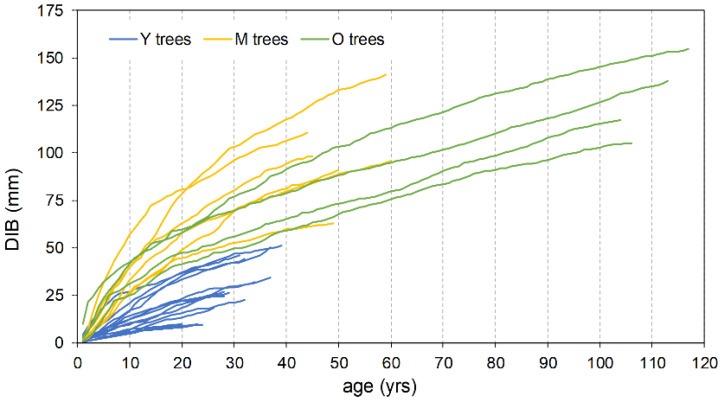
Cumulative DIB increment vs. age for trees with cores reaching the pith (n = 25).

**Figure 7 plants-13-00061-f007:**
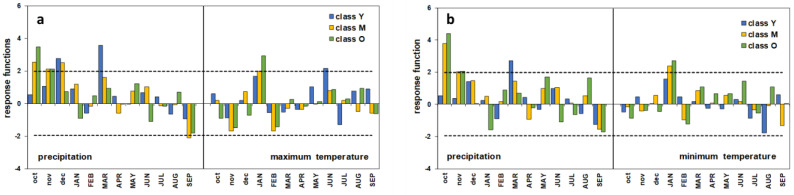
Response functions of old (O), mature (M), and young (Y) trees. (**a**) P–Tmax relationship; (**b**) P–Tmin relationship. Horizontal dashed lines correspond to 95% significance threshold of 1000 bootstrap replications; months code in lowercase represents previous year’s months, months code in capitals represents current year’s months.

**Table 1 plants-13-00061-t001:** Main biometric data of young (Y), mature (M), and old (O) trees with standard deviation (std) and coefficient of variation (CV). DBH: average diameter at breast height; H: average tree total height; CW: average crown width area; CR: average crown radius.

Age Class		N Trees	DBH			H			CW			CR		
			(cm)	Std	CV (%)	(m)	Std	CV (%)	(m^2^)	Std	CV (%)	(m)	Std	CV (%)
Y	≤40	56	12.56	6.61	52.6	5.52	2.09	37.9	9.64	7.76	80.5	1.62	0.72	44.4
M	41–80	31	27.61	6.44	23.3	9.53	1.62	17.0	32.91	13.72	41.7	3.22	0.7	21.7
O	>80	24	42.22	5.13	12.2	14.76	1.27	8.6	62.44	18.52	29.7	4.42	0.68	15.4
total		111												

**Table 2 plants-13-00061-t002:** Descriptive statistics and response function data of young (Y), mature (M), and old (O) trees. MRW: mean ring width over the entire series length; MSm: mean sensitivity of master series; MSi: average of mean sensitivity of the individual series; CC: cross-dating coefficient; CM: correlation of individual series with master; AC_1_: autocorrelation coefficient of order 1 on raw and standardized (std) series; R_V_: correlation coefficient on verification data set; r/s: ratio of correlation coefficient and its standard deviation; ns: non-significant.

		Descriptive Statistics of Ring Width									Response Function						
Age Class		N Trees (N Cores)	Max Time Span at DBH		Mean Sensitivity						N Trees	P–Tmax			P–Tmin		
				MRW	MSm	MSi	CC	CM	rawAC_1_	stdAC_1_		R_V_	r/s	*p*	R_V_	r/s	*p*
Y	≤40	52 (98)	1958–1997	1.310	0.292	0.412	0.709	0.674	0.610	0.042	26	0.80	1.85	<0.1	0.79	0.89	ns
M	41–80	28 (53)	1932–1997	1.850	0.298	0.355	0.839	0.737	0.754	0.011	23	0.77	2.03	<0.05	0.75	1.68	<0.1
O	>80	23 (45)	1865–1997	1.210	0.286	0.394	0.726	0.700	0.767	0.016	21	0.73	2.13	<0.05	0.76	2.01	<0.05
	total	103 (196)									70						

## Data Availability

Data are contained within the article and Appendix A.

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
