# Peer review of "Differential Tree Growth Response to Management History and Climate in Multi-Aged Stands of Pinus pinea L."

_plants, 2023, doi:10.3390/plants13010061_

Round 1

Reviewer 1 Report

Comments and Suggestions for Authors

This is a well-researched and well-written manuscript that helps to fill a gap in our understanding of stone pine dendroecology and stand management. The methodology used is appropriate and have been carefully applied. The use of core analysis to select the tree age categories is robust.

My only suggestion to improve the manuscript is to remove the statement on root system development from the abstract as no new data were acquired in this field in the research that was undertaken. Also, it is surprising that young trees (up to 40 years of age) do not access the groundwater aquifer (line 552) in the dry season. This could be easily checked by excavating a few trees in this age group.

Please check for minor errors such as: Cistus ssp.; citing authorities for taxa when first cited (e.g. line 74).

I find the change in the common name from stone pine in the abstract to umbrella pine in the body of the paper perplexing – please use one name throughout.

Author Response

Reviewer: This is a well-researched and well-written manuscript that helps to fill a gap in our understanding of stone pine dendroecology and stand management. The methodology used is appropriate and have been carefully applied. The use of core analysis to select the tree age categories is robust.

Authors: We thank a lot the reviewer for his appreciation of our manuscript, that let us feel happy. We include below a point-by-point response to his comments and the corresponding revisions/corrections highlighted/in track changes in the re-submitted files.

Reviewer: My only suggestion to improve the manuscript is to remove the statement on root system development from the abstract as no new data were acquired in this field in the research that was undertaken.

Authors: Actually all our statements on root system development of younger tree, both in the abstract and in the main text, are but assumptions based on existing literature aimed at providing an explanation of the outcomes of our research. We prefer not to remove this statement from the abstract in order not to weaken the following sentence, pointing that “appropriate and timely thinning could be a suitable managing approach to promote sustained growth rates (here intended as the global tree growth at the above and below ground level, as specified in the main text), as well as healthy and vital trees at all life stages.”

Reviewer: Also, it is surprising that young trees (up to 40 years of age) do not access the groundwater aquifer (line 552) in the dry season. This could be easily checked by excavating a few trees in this age group.

Authors: Yes, we can agree with the reviewer that this issue could raise some surprise, and checking it by excavating a few trees could be very interesting either to evaluate the real root development in the different age classes or to measure the true depth of the aquifer in the investigated area, especially in the light of the current climate changes. However, to concretely carry out excavations should not be so simple or easy, particularly considering that the investigated stands are included within a protected area with severe restrictions about any kind of manipulations against the natural and biological assets. Moreover, our hypotheses, though speculative, can be considered quite sound on the basis of outcomes from some literature reporting data on the depth of umbrella pine root system (e.g. Cutini, A.; Chianucci, F.; Manetti, M.C. Allometric relationships for volume and biomass for stone pine (Pinus pinea L.) in Italian coastal stands. iForest 2013, 6, 6, 331-335. doi: 10.3832/ifor0941-006), or on the importance of a long taproot of umbrella pine to avoid competition from understory vegetation, particularly during the juvenile phase of stand development (e.g. Frattegiani, M.; Mencuccini, M.; Mercurio, R.; Profili, W. Quantitative analysis of Stone pine (Pinus pinea L.) root systems morphology and its relationships with water table and soil characters. Inv. Agrar. 1994, Fuera de Serie 3, 405–41, and Mazza, G.; Sarris, D. Identifying the full spectrum of climatic signals controlling a tree species' growth and adaptation to climate change. Ecol. Indic. 2021, 130, https://doi.org/10.1016/j.ecolind.2021.108109). To highlight the issue, we added the citations Cutini et al., 2013 at line 552 (line 558 in the current .word re-submitted version) and Frattegiani et al., 1994 (actually already listed in the references, but omitted by distraction in the main text) at line 553 (line 559 in the current .word re-submitted version).

 Reviewer: Please check for minor errors such as: Cistus ssp.; citing authorities for taxa when first cited (e.g. line 74).

Authors: As suggested, we checked the text for these minor errors. Concerning the suggestion related to line 74, we reported (in brackets) the official definition of the respective habitat as listed in the ‘Habitats’ Directive 92/43 EEC, so we think it is correct not to add the authorities.

Reviewer: I find the change in the common name from stone pine in the abstract to umbrella pine in the body of the paper perplexing – please use one name throughout.

Authors: As suggested, we corrected it.

Reviewer 2 Report

Comments and Suggestions for Authors

The submitted manuscript deals with interesting topic of the growth performance in multi-aged Pinus pinea forest stand. The uneven-aged studied stand therefore represents a valuable opportunity to study age-related processes. It is based on the dendrochronological evaluation of three basic groups of trees old (O), mature (M) and young (Y). The authors grouped the trees a priori into three classes. It seems that this forest stand has three different generations. I would rather expect a continuous regeneration of trees from different age categories and to group a posteriori the sample trees into classes after evaluating increment cores. May be there will not occur significant differences, but I would consider this process as more appropriate, since there is rather high heterogeneity in age distribution within groups M and Y. 

I agree with the statement that the oldest trees originated from afforestation and thus more or less even-aged one-storey stand. Later on due to additional planting and natural regeneration the stand became multi-aged structure.

I consider the methods used as standard ones and properly used. There have appeared in the manuscript several typographical imperfections they need improvements. They are listed in the attached file.

Author Response

Reviewer: The submitted manuscript deals with interesting topic of the growth performance in multi-aged Pinus pinea forest stand. The uneven-aged studied stand therefore represents a valuable opportunity to study age-related processes. It is based on the dendrochronological evaluation of three basic groups of trees old (O), mature (M) and young (Y).

Authors: We thank a lot the reviewer for his appreciation of our manuscript, that let us feel happy. We include below a point-by-point response to his comments and the corresponding revisions/corrections highlighted/in track changes in the re-submitted files.

Reviewer: The authors grouped the trees a priori into three classes. It seems that this forest stand has three different generations. I would rather expect a continuous regeneration of trees from different age categories and to group a posteriori the sample trees into classes after evaluating increment cores. May be there will not occur significant differences, but I would consider this process as more appropriate, since there is rather high heterogeneity in age distribution within groups M and Y.

Authors: Truly, the reviewer is completely right when saying he should expect a continuous regeneration of trees from different age categories, because this is the real pattern issuing from our data. However, in order to be able to analyse the processes related to the goals of the investigation (see lines 80-87, but lines 87-93 in the current .word re-submitted version) we necessarily had to establish the thresholds discriminating age classes from each other according to a number of criteria. As reported in Materials and Methods line 234 (lines 242-243 in the current .word re-submitted version), in fact the first grouping criterion was “the age distribution patterns of all trees of the investigated stand”, involving the ring counting (and width measuring for the purpose of the correct attribution of age) from all sampled increment cores. We additionally considered (lines 235-239, but lines 243-247 in the current .word re-submitted version): 1) the longevity of the species and the age of the oldest sampled individuals, 2) the age of “useful” fructification (over 40 years) with its implications in terms of the traditional function of pine nuts production and the inherent silvicultural management, and 3) some dendrochronological issues already investigated in a monography on Pinus pinea focusing on methodological aspects of dendrochronological researches. As a matter of fact, the most debated age breakdown for us was the one between classes Y and M, because for the purposes of dendrochronological analysis in order to achieve a greater robustness of statistical significance a series length of at least 40 years is desirable. Therefore, all these conditions should satisfy at a time the need to obtain both a sufficient length and representation of each class, and a good compromise between a reasonable disaggregation of the data, possibly taking into account also crucial moments of the stand management history. According to the above, we are in the opinion that actually the expectations of the reviewer can be considered as fulfilled.

Reviewer: I agree with the statement that the oldest trees originated from afforestation and thus more or less even-aged one-storey stand. Later on due to additional planting and natural regeneration the stand became multi-aged structure.

Authors: Thank you for sharing your opinion, which sustain us about the soundness of our interpretation of the data.

Reviewer: I consider the methods used as standard ones and properly used. There have appeared in the manuscript several typographical imperfections they need improvements. They are listed in the attached file.

Authors: Thank you for such a careful revision of our manuscript, allowing us to remediate to these typographical imperfection. However, I wish to notify that some of them (e.g. subscripts) were correctly typed in the .word draft version of the manuscript, but imperfections arose during the automatic conversion process from the .word to the .pdf file after the submission. This is also the case of Table 2, where the font size of the the parameters MRW and AC1 is consistent to stay into a line in the .word version of the manuscript. We nevertheless checked all of them and incorporated some corrections where necessary.